# Transcriptional Dynamics Underlying Somatic Embryogenesis in *Coffea canephora*

**DOI:** 10.3390/plants14071108

**Published:** 2025-04-02

**Authors:** Marcos-David Couoh-Cauich, Hugo A. Méndez-Hernández, Rosa M. Galaz-Ávalos, Ana Odetth Quintana-Escobar, Enrique Ibarra-Laclette, Víctor M. Loyola-Vargas

**Affiliations:** 1Unidad de Biología Integrativa, Centro de Investigación Científica de Yucatán, Calle 43, No. 130 x 32 y 34, Mérida 97205, Yucatán, Mexico; md.couoh@gmail.com (M.-D.C.-C.); hugomendez2206@gmail.com (H.A.M.-H.); gaar@cicy.mx (R.M.G.-Á.); odetth@gmail.com (A.O.Q.-E.); 2Red de Estudios Moleculares Avanzados, Instituto de Ecología, A.C. (INECOL), Xalapa 91073, Veracruz, Mexico; enrique.ibarra@inecol.mx

**Keywords:** auxins, cytokinins, transcriptome

## Abstract

In *Coffea canephora*, a direct somatic embryogenesis (SE) protocol has been established by pretreating plants with NAA and kinetin, followed by the induction of leaf explants in a liquid medium with BA. Through a transcriptomic analysis of 10 key moments of the induction of SE in *C. canephora*, we were able to establish that the transcriptional responses of this process are divided into four stages. These stages correspond to the pretreatment, characterized by the positive regulation of genes associated with cell wall remodeling, flavonoid biosynthesis, and antioxidant activity that prepare the explants for the intense cellular activity that represents the induction of SE. During the first few hours, the early response to induction occurs, characterized by the highest number of differentially expressed genes, most related to the response to multiple stimuli. At 24 h, a late response begins with the upregulation of genes related to energy production and the biosynthesis of amino acids. Finally, *WOX*, *BBM*, and *ABI3* genes are upregulated during the embryogenic response. The downregulation of genes related to the circadian cycle, photomorphogenesis, photosynthesis, and chloroplast components were observed throughout the process. The detailed analysis of the primary transcriptional responses that occur during the SE of *C. canephora* helps us to clarify how auxins and cytokinins orchestrate the integration of different networks of plant metabolism and lead to the development of somatic embryos.

## 1. Introduction

Plant somatic cells can form embryos under certain conditions or appropriate stimuli, known as somatic embryogenesis (SE). SE shares similarities with zygotic embryogenesis (ZE). Still, the possibility of manipulating the conditions under which this event occurs in vitro has positioned it as a model system for studying cell differentiation and embryo development in plants [1]. This experimental model has allowed significant advances in areas such as the propagation and conservation of genetic resources of endangered species [2,3,4], as well as the development of transgenic cotton plants [5], maize [6], and tobacco [7], highlighting their value in biotechnological applications [1].

SE has also become a valuable instrument for basic studies of SE induction and somatic embryo development [8]. Huge advances have been made since it was discovered that the SE process that occurs in nature, mainly in plants of the Kalanchoe genus species [9,10], could be reproduced in the laboratory [11]. Important mechanisms involved in the change from somatic cells to embryogenic cells are still unknown, and the coffee plant is no exception.

Plant growth regulators (PGRs) regulate plants’ regeneration ability to influence the somatic embryo fate [12]. Since the pioneering work of Skoog and Miller [13], which established that the balance between auxins and cytokinins (CKs) regulates root and shoot formation in plant tissue culture, numerous studies have explored the role of PGRs, particularly auxins, and CKs in plant development and regeneration [12,14,15]. Auxins and CKs have also been found to be essential in developing SE. The synthesis of indole-3-acetic acid (IAA) is crucial for the induction of SE [16,17,18]. The inhibition of IAA biosynthesis or its transport turns off the induction of SE [19,20].

The first SE reports were made in carrots in the 1950s [11]. Hundreds of species have successfully produced somatic embryos in vitro. The development of efficient SE protocols in species such as *A. thaliana*, carrot, cotton, and alfalfa has allowed them to be used as model systems for the study of the mechanisms that lead to the reprogramming of somatic cells to embryos [21].

Coffee is an essential crop for the economy of many countries. Within the *Coffea* genus, *C. arabica* and *C. canephora* together represent 100% of the world’s coffee production, which has influenced the development of numerous SE protocols for the clonal multiplication of elite genotypes or heterozygous hybrids [22,23,24]. The SE objectives for Coffee spp. have been mainly economic. However, there has been a significant advance in understanding the molecular mechanisms that occur during coffee’s SE. Loyola-Vargas et al. [22] and Etienne et al. [24] compile the advances achieved through decades of studying the SE of *Coffea* spp.

The two coffee species, *Coffea arabica* and *C. canephora*, used for the production of the beverage, have been the subject of studies for propagation by SE since the pioneering work of Staritsky [25] and Söndahl and Sharp [26]. Today, there are protocols for obtaining somatic embryos in both species [22]. These protocols are constantly being modified to improve them and scaled up industrially [27,28,29,30]. The coffee plant, like a large number of species, also presents the problem that different species of the same genus, and even varieties of the same species, respond differently to the induction of SE or, in other words, require different conditions for the induction of SE [28,31,32,33].

For *C. arabica*, the indirect SE process developed by Etienne [34] is one of the most commonly studied; the multiple stages that make up this protocol have been characterized and analyzed from the metabolomic and transcriptomic approach, managing to identify key marker genes and metabolites for each of the stages [35,36].

In the case of *C. canephora*, the protocol described by Quiroz-Figueroa et al. [37] has used omics to explore the changes in the transcriptome globally [38] and proteome [39,40] during SE. In particular, the role of the *YUCCA* [20], *GH3* [41], *ARF*, and *AUX/IAA* [38] gene families has been analyzed. The accumulation pattern of free auxins and their conjugates has also been identified [42], as well as the distribution of free IAA and *PIN* transporters throughout the entire process [43]. Additionally, the role of CK homeostasis during SE in *C. canephora* has been explored [44], as well as the role of epigenetics in regulating the expression of key TFs such as *LEC1*, *BBM*, and *WOX4* [45].

This study aims to contribute to understanding the SE process using a transcriptomic approach. We analyzed how auxins and cytokinins orchestrate the integration of different networks of plant metabolism and lead to the development of somatic embryos.

## 2. Results and Discussion

### 2.1. SE of C. canephora as a Study Model

Somatic embryogenesis in *C. canephora* has been widely studied in our laboratory. This process consists of two stages. First, plants with approximately six pairs of leaves in MS medium without growth regulators are transferred to a medium with naphthaleneacetic acid (NAA) and kinetin (KIN), which are maintained for 14 days under photoperiod conditions. This pretreatment stage is essential for the induction of SE because it reduces the time and yield of embryo formation; without it, the embryogenic response is drastically reduced [46]. Subsequently, leaf discs are cut, exposed to a modified Yasuda liquid medium with 6-benzyladenine (BA) [47], and grown in dark conditions. It is in this medium that embryogenic structures begin to form and develop. During the first 14 days, the explants showed an apparent lack of a response (Figure 1B–F); however, there are many internal changes in different components of the tissues [41,42,44]. The formation of a proembryogenic mass can be observed at the edges of the explants after 21 days (Figure 1G). The first globular structures can be observed on day 28 after the induction of SE (Figure 1H). SE induction ends before embryo formation at the heart stage, when embryo differentiation begins [48]. Before embryo formation, cell proliferation is observed, which gives way to the proembryogenic mass where the meristematic centers and, later, the somatic embryos are formed [45].

During the pretreatment stage, NAA is transported to the leaves and remains constant through the SE induction process; as a response, an increase in free and conjugated IAA levels is observed, reaching their maximum on day 0; this increase is not steady. A decrease in total IAA is observed at 9 days before induction (DBI) [42]. The incubation of seedlings with 3-^14^C-tryptophan demonstrated that the increase in IAA during pre-treatment arises from Trp-dependent *de novo* biosynthesis and not from conjugate hydrolysis [20].

In *A. thaliana*, it has been shown that all tissues of young seedlings can biosynthesize IAA; this capacity is more significant in those tissues with active cell division than those growing due to expansion [49]. Negative regulation of the *TAR2*, *YUC1*, *YUC2*, *YUC4*, and *YUC6* genes has also been observed in response to the exogenous application of synthetic auxins such as 2,4-dichlorophenoxyacetic acid, accompanied by a reduction in endogenous IAA levels [50]. Exogenous application of NAA at less than 0.1 µM promoted an increase in the number of cells in *A. thaliana* seedlings, while at 0.4 µM, the number of cells was reduced, but their size increased. In addition, the use of DR5::GUS allowed Kalve et al. [51] to observe that the auxin response mainly occurred in proliferating leaves compared to fully developed cotyledons, suggesting that NAA is transported acropetally to the leaves, thus inducing their response to auxin.

### 2.2. Global Transcriptomic Analysis of C. canephora SE

To understand the transcriptional regulation in *C. canephora* during the SE induction process, RNA sequencing (RNA-seq) of key moments was performed. This analysis included samples collected at different time points, from the pretreatment stage (14 days before induction) to 28 days after induction, when the formation of somatic embryos was more evident.

A total of 1,198,039,799 clean reads were obtained via RNA-seq sequencing, corresponding to 20 *C. canephora* libraries at key times of embryogenesis induction (Appendix A). The average was approximately 64,396,326 reads per library, except for samples corresponding to 28 days after induction (DAI), sequenced at a lower depth (19,452,962 reads on average). The percentage of clean reads mapping to the reference genome ranged from 87.03% to 91.23%, with an average of 89.4% (Appendix A). Despite the observed variation in the sequencing depth, particularly noticeable in the raw counts (Appendix A), the analysis of log_2_(TPM+1)-transformed data (Appendix A) revealed no substantial differences between the libraries. This suggests that the data normalization process effectively mitigated the impact of sequencing depth disparities.

Variance stabilization transformation (VST) was applied to the raw counts for correlation analysis. This analysis shows a Pearson correlation coefficient more significant than 93% for all biological replicates (Appendix A). The high Pearson correlation coefficient between biological replicates indicates a strong consistency in gene expression patterns between samples from the same experimental group, providing a solid basis for further analyses.

Principal component analysis (PCA) reveals distinct patterns in gene expression throughout the embryogenesis process. The first two principal components (PC1 and PC2) explain 34.61% and 23.73% of the total variability, respectively (Figure 2A). Most replicates for each sample cluster closely indicate high reproducibility, except samples 9DBI and 0DBI, which show a higher dispersion between replicates.

The PCA plot reveals four distinct clusters corresponding to critical developmental stages. The first cluster groups the pretreatment samples (14DBI, 9DBI, and 0DBI). The second cluster, located at the bottom of the plot, includes samples from the first and second hours after induction (1HAI and 2HAI), representing the early induction stage. The third cluster, which comprises the first and third days after induction, corresponds to the middle induction stage. Finally, the fourth cluster, positioned on the right side of the plot, includes samples from weeks 2, 3, and 4 post-induction (14DAI, 21DAI, and 28DAI). At this later stage, the tissue shows clear signs of embryogenic structure and the formation and development of embryogenic structures. This spatial distribution of samples in the PCA reflects the temporal progression of the embryogenesis process and suggests significant transcriptomic changes between the different stages.

A PERMANOVA test confirmed significant transcriptomic differences between developmental stages (*R*^2^ = 0.83, *p* = 0.001), reinforcing the patterns observed in the PCA and indicating that they reflect true biological variation.

The new assembly comprises 28,852 protein-coding genes, of which 27,853 (96.54%) were successfully assigned to specific chromosomes. Only 999 genes could not be transferred to any of the chromosomes. Two complementary strategies were used for the functional annotation of these genes: a BLAST analysis using the *A. thaliana* proteome as a database, which allowed the annotation of 23,378 genes, and an analysis with InterProScan using the PANTHER and Pfam databases, which resulted in the annotation of 26,766 genes. Together, these approaches allowed the functional annotation of 26,989 proteins, equivalent to 93.54% of the total coding genes identified.

Differential gene expression analysis was conducted using the DESeq2 package, comparing each time point in the transcriptome to the reference time point, 14DBI (14 days before induction), corresponding to a *C. canephora* plant grown in vitro (Figure 1A). Across all time points sampled during the SE process, 13,119 genes were identified as differentially expressed in at least one point.

Our work did not identify the upregulation of *TAA* and *YUC* family genes encoding enzymes of IAA biosynthesis. Transcriptome analysis of the first pair of true leaves of NAA-treated *A. thaliana* seedlings revealed that *TAA1*, *YUC2*, and *AUX1* were upregulated, while neither *GH3* nor *PIN* showed statistically significant changes [51]. In contrast, we observed that during *C. canephora* pretreatment, no genes of the *TAA*/*TAR*, *YUC*, or *GH3* family showed statistically significant changes (*p*-adjusted < 0.05), while homologs of *PIN1* and *PILS5* were upregulated. However, this study used the second and third pair of *C. canephora* leaves, where the response to exogenous auxin could be lower. Additionally, we observed that auxin response genes *ARF4*, *IAA19*, and *IAA27* were upregulated, while *ARF19* and *IAA13* were downregulated.

Figure 2B illustrates the differentially expressed genes (DEGs) throughout the SE induction process. More upregulated genes are generally observed at all time points than downregulated genes. The pretreatment times 9DBI and 0DBI exhibited the lowest number of DEGs (640 and 250 upregulated genes and 1003 and 189 downregulated genes, respectively).

A marked increase in DEGs was observed during the early hours of induction, peaking at 2 h after induction (2HAI) with 3500 upregulated and 3103 downregulated genes. This was followed by a decline in DEGs at 24 h (1 DAI) and 72 h (3 DAI). Starting 14 days after induction (14DAI), the number of DEGs began to rise again, continuing to increase until reaching the highest number of DEGs at 28DAI, with 4402 upregulated and 3435 downregulated genes. The gene enrichment analysis of the DEGs identified at different points in the SE process of *C. canephora* provides an overview of gene regulation, allowing the identification of specific biological processes and metabolic pathways that are altered during the transition from somatic to embryogenic tissue.

Representatively, Figure 2C–E shows GO terms for biological processes and cellular components (molecular functions are shown in Appendix A), as well as KEGG pathways enriched with lower *p*-values at each stage of the process. A complete list of enriched terms is provided in Appendix A.

In general, it was observed that terms associated with the light stimulus, photosynthesis, chloroplast components, organization, and the circadian cycle were enriched in genes downregulated throughout the process. After the pretreatment, induction occurs under dark conditions; however, some responses associated with light stimuli are already downregulated at 9DBI.

The effects of light on SE and organogenesis have been widely studied; the light regime and the light spectrum used can directly affect the embryogenic response [52]. In this work, we identified several genes involved in light signaling, circadian cycle, and photosynthesis that were downregulated, with *LHY* being differentially regulated from 9DBI, while *PIF1*, *PIF3*, and *PIF4* were from 0DBI. The promoter region of *LHY* has been shown to contain an ARF-binding site [53]. On the other hand, the *PIF* family of TFs promotes growth in the dark by binding to the promoters of hundreds of genes. However, the expression of some members, such as *PIF1* and *PIF3*, do not change during the day, so they are not controlled by the circadian cycle, while the expression of *PIF4* and *PIF5* oscillate during the day [54,55,56]. Auxin biosynthesis and response genes such as *YUC1*, *YUC2*, and *YUC6* and *ARF5*, *ARF8*, and *ARF16* are induced by inhibiting *PIF4* [57]. Additionally, it has been determined that circadian clock key genes (*LHY*, *GI*, and *ELF4*), as well as genes involved in photosynthesis (*PORA* and *CAB1*) in indirect SE of *C. arabica*, remain downregulated throughout the process [36].

The analysis of genes with reduced expression at 9DBI within the category GO:0006355~regulation of DNA-templated transcription allowed us to identify a regulatory network whose homologs in *A. thaliana* are associated with key processes such as the circadian cycle, light signaling, and chloroplast development. Among the identified genes, *LHY* (AT1G01060), *LNK1*, and *LNK2* (AT5G64170 and AT3G54500) stand out, as well as the transcription factors of the BBX family, including *BBX19*, *BBX20*, *BBX24*, and *BBX29* (AT4G38960, AT4G39070, AT1G06040, and AT5G54470, respectively). Homologs of *RVE1* and *RVE8* (AT5G17300 and AT3G09600) and *CDF2* and *CDF3* (AT5G39660 and AT3G47500) were also detected, together with regulators of auxin metabolism such as *ARF19* and *IAA13*.

In contrast, responses to different stimuli were found to be upregulated throughout the SE process. However, it is during the 2HAI, 1HAI, and 0DBI induction that a more significant enrichment of terms related to response mechanisms is observed, with 23, 18, and 17 terms, respectively, found with responses to different types of stimuli (all terms associated with the various types of response can be seen in Appendix A). Some of the terms that are upregulated include “response to wounding” (GO:0009611), “pattern recognition receptor signaling pathway” (GO:0002221), or “defense response to other organisms” (GO:0098542).

Both the pectin catabolic process (GO:0045490) and cell wall modification (GO:0042545) were enriched among the differentially expressed genes (DEGs) in the 9DBI-Up, 3DAI-Up, 14DAI-Up, 21DAI-Up, and 28DAI-Up conditions. Some genes are expressed in early stages (9DBI and 0DBI), such as *XTH22* (AT4G16563), which encodes a xyloglucan protein endotransglucosylase/hydrolase; several members of pectinesterases and their inhibitors, such as *PME12* (AT2G45220), *PME17* (AT1G11580), *PME18* (AT3G05620), *PME22* (AT5G27870), *PME28* (AT3G47400), *PME33* (AT4G02320), and *PME40* (AT2G21660); genes encoding proteases, such as *F15K9.16* (AT1G03220), *F15K9.17* (AT2G16250), *F17M5.250* (AT2G34930), and *T7O23.17* (AT5G57560), belonging to the peptidase A1 family. The overexpression of genes from the peroxidase family, such as *PER12*, was found (AT5G17820), including *PER57* (AT5G64120) and *PER71* (AT2G26440); also found were genes belonging to the *SKU5* family, in particular *SKS4* (AT1G74000), *SKS5* (AT1G33590), and *SKS6* (AT3G55430). Finally, genes *EXLA2* (AT4G08950) and *EXPA6* (AT1G03230) were found to be associated with the expansin family, some of which stimulate acid growth, and *EXO* (AT2G28950) was involved in signaling processes. Together, these genes could participate in tissue growth by relaxing and remodeling the wall induced by pretreatment.

Genes associated with flavonoid biosynthesis from the *LEA* family, several chaperones from different subfamilies, and genes related to oxidative stress overexpressed during 9DBI and 0DBI were also identified. The complete list of genes involved in these processes is presented in Appendix A.

The terms associated with the response to the different growth regulators show an enrichment during 1HAI and 2HAI but not throughout the process. The response to some growth regulators shows downregulation, as in the case of the “auxin response” (GO:0009733). In contrast, “response to abscisic acid” (GO:0009737) shows upregulation, which could indicate an adjustment of specific growth regulators during the early stages of embryogenic induction.

On the other hand, in the days after induction, starting at 1DAI and 3DAI, there is a notable overexpression of genes associated with ribosome components and primary metabolism. Activation of genes related to energy production during the “glycolytic process” (GO:0006096), “tricarboxylic acid cycle” (GO:0006099 and ath00020), and “oxidative phosphorylation” (ath00190) is observed. The overexpression of these genes is maintained until 28DAI. Appendix A shows the carbon metabolism pathways to which the DEGs belong, where the genes involved in glycolysis and the TCA cycle are upregulated while the genes involved in carbon fixation due to the Calvin cycle and starch biosynthesis are under-expressed. The TCA pathway and oxidative phosphorylation take place in mitochondria, and the terms associated with this organelle are shown in Appendix A, where it can be observed that most are overexpressed from 1HAI.

This metabolic reconfiguration implies that cells adapt to meet the heightened energy demands associated with SE. In this context, cells are likely utilizing stored energy reserves or capitalizing on carbon sources available in the culture medium to satisfy these elevated metabolic requirements. This metabolic adjustment appears crucial for sustaining the intense cellular activity characteristic of SE, particularly in cell division and differentiation. Additionally, the sustained activation of defense mechanisms and the modulation of responses to growth regulators suggest that SE in *C. canephora* involves a complex signaling network that integrates stress responses with developmental regulation mechanisms.

In leaf explants, somatic embryos can originate from dedifferentiated mesophyll cells or the vascular procambium. Molecules released from the wound site can induce cell dedifferentiation and SE [58,59]. Reactive oxygen species (ROS) production begins within seconds of wounding in *M. truncatula* leaf explants [58,60]. ROS production is essential for the dedifferentiation and proliferation of *A. thaliana* protoplasts [61]. Optimum modulation of ROS through gene upregulation during SE has been demonstrated in *M. truncatula* and cotton [62,63].

A comparison with other SE studies in various plant species shows that some GO (Gene Ontology) categories are recurrently enriched during this process. For example, in *Z. mays*, enrichment has been observed for processes related to the oxidative stress response (GO:0006979), PGR-mediated signaling (GO:0009755), positive regulation of transcription, the citric acid cycle, phenylpropanoid biosynthesis, amino acid synthesis, and glycolysis- and gluconeogenesis-associated metabolic pathways [64]. In *A. thaliana*, prominent GO categories include carbohydrate, lipid, and amino acid metabolic processes, hormone biosynthesis, metabolism, and cell wall organization and biogenesis [65]. Similarly, in *C. arabica*, pathways related to hormonal signaling (mainly auxins and cytokinins), metabolic processes (carbohydrates, starch, proteins, and secondary metabolism), regulatory pathways (the regulation of gene expression, pattern specification, and embryonic development), responses to stress and wounding, processes related to mitosis (cell cycle, cell division, and cell wall and chromatin organization), and photosynthetic functions such as photosynthesis and circadian rhythm have been identified [36]. This work suggests that in vitro SE is controlled by conserved regulatory networks and species-specific variations.

### 2.3. Identification of Stage-Specific Coexpression Networks Associated with SE of C. canephora

A gene coexpression analysis was performed using Weighted Gene Co-expression Network Analysis (WGCNA) to identify modules of genes with similar expression patterns during SE in *C. canephora*. Log_2_ (TPM+1) values were used to construct an “unsigned” coexpression network. Scale independence and mean connectivity were evaluated to determine the optimal network threshold. A value of 18 was selected as the optimal threshold based on the scale independence and mean connectivity graphs presented in Appendix A. Scale independence refers to the extent to which the network topology resembles a scale-free network, a typical property in robust biological networks. The scale independence graph shows that starting from a value of 18, the R^2^ index of fit to a scale-free network stabilizes near 0.8, indicating a good approximation to this property. On the other hand, the mean connectivity decreases rapidly until it reaches a low and stable value around the threshold of 18.

Appendix A presents the dendrogram from the hierarchical clustering analysis applied to the genes and the colors assigned to each module. A total of 26 modules were identified, plus the gray module, where genes that did not group into any other module were clustered. Each module is represented by a distinct color in the dendrogram’s lower bar, with the tree’s main branches representing groups of genes with highly correlated expression patterns. The complete set of genes included in this analysis, their normalized log_2_(TPM+1) values, and the module to which they belong can be found in Appendix A.

Two correlation analyses were performed using Pearson’s coefficient to identify coexpression modules associated with specific stages of SE. The first examined the correlations between the modules and the process’s 10 time points. In contrast, the second grouped these points into four visible phenotypic stages: pretreatment, early response, late response, and embryogenic response.

The obtained correlation coefficients (r) range from −1 to 1, indicating positive or negative relationships between the modules and the stages of the process. The statistical significance of these relationships was evaluated by calculating the associated *p*-values using Student’s t-distribution. Correlations with r > 0.75 were considered strong.

Both analyses provided complementary perspectives, revealing modules with significant correlations at specific moments or stages spanning several process points. Figure 3A illustrates the various modules obtained, the number of genes in each, and their relationship with the four stages of SE in *C. canephora*. The turquoise module has 1538 genes, while the dark orange is the smallest, with 82 genes. Modules with fewer than 50 genes were not considered.

Distinctive correlations were observed: the midnight blue module shows a strong positive correlation with the pretreatment stage, the orange and red modules are mainly associated with the early response, the black module is linked to the late response, and the brown and turquoise modules correlate with the embryogenic response.

It is important to note that not all time points within each process stage correlate uniformly. Appendix A details the modules’ correlation with the 10 studied time points, revealing interesting variations. For example, midnight blue exhibits a stronger correlation with 14DBI compared to 9DBI and 0DBI, although the direction of the relationship is maintained. This pattern is repeated in other modules, with the case of dark red, which shows a strong correlation with 2HAI but a negative correlation with 1HAI.

Modules with correlations r > 0.75 were selected for a more detailed analysis. Two types of graphical representations were generated to examine the expression patterns of each module. Heatmaps with hierarchical clustering offer a global view of gene expression patterns throughout the SE process. Different color scales were applied: values < 1 are represented in a gradient from white to light blue, with white indicating genes with 0 Transcripts Per Million (TPM), interpretable as low-expression or silenced genes. The second type of graph plots the average expression pattern of the module, facilitating the identification of significant changes at critical moments of the process. Both visualizations allow the identification of modules whose activity varies at key points, suggesting their possible involvement in specific biological events, such as the induction or development of somatic embryos.

Figure 3B and Appendix A focus on three modules highly correlated with specific process stages: brown, black, and red, respectively. Each panel includes a heat map showing individual gene expression patterns, accompanied by a line graph illustrating the average expression trajectory of the module over time.

The black module exhibits higher gene expression at 1DAI and 3DAI than other time points. In the brown module, numerous genes show low expression or silencing from 14DBI to 3DAI, with higher expression levels from 14DAI to 28DAI. The red module presents its maximum expression level at 1HAI and 2HAI.

Other modules with strong correlations are presented in Appendix A. The dark red module shows higher expression at 2HAI but not at 1HAI. In grey60, genes experience downregulation at 1HAI and 2HAI, followed by higher expression at 1DAI and 3DAI. Light cyan follows a pattern similar to dark red, while genes belonging to the midnight blue module exhibit the highest expression values during pretreatment. The orange module presents a pattern similar to the red module. Finally, the turquoise module contains genes with expression patterns similar to the brown module but with an increase in expression from 1DAI that is maintained until the end of the process.

An unsigned gene coexpression network does not differentiate between positive or negative correlations. Therefore, genes with correlated expression can be grouped in the same module, whether overexpressed or repressed at the same process stages. This is particularly useful when identifying sets of genes that respond in a coordinated manner to specific stimuli, regardless of whether their response is in the opposite direction. For modules strongly correlated with a particular stage of the SE process, a k-means analysis was performed to identify more homogeneous subgroups based on their expression pattern. This facilitates a more detailed analysis of the functionality of these subgroups and allows for the evaluation of their possible gene enrichment, offering a more precise view of the contribution of these genes to the somatic embryogenesis process.

The red module is strongly associated with the early response to SE induction. This module is divided into three distinctive clusters, which can be observed in Appendix A, each with particular expression patterns and biological functions. Genes with high expression characterize the first cluster during 1HAI and 2HAI, contrasting with low levels in the other stages. This group presents an enrichment related to the response to limited oxygen levels. Regarding molecular functions, a significant enrichment is observed in activities associated with transcription factors. Unlike the first group, the second cluster also exhibits upregulation at 1HAI and 2HAI but maintains considerable expression in the previous and subsequent stages. The enriched terms in this cluster are linked to responses to various stimuli, including mechanical injuries and growth regulators. The third cluster presents an inverse dynamic to the previous two: the genes in this group experience downregulation at 1HAI and 2HAI. This cluster shows an enrichment related to DNA packaging and protein processing. In this initial stage of SE, the activation of transcription factors, stress response pathways, and the modulation of DNA-related processes appear crucial.

The black module is strongly associated with the late response to SE induction. This module comprises three main clusters, each with distinctive functions and expression patterns represented in Appendix A—cluster 1 shows enrichment in biosynthesis processes and macromolecule metabolism, including ribosome biogenesis. Cluster 3 exhibits an enrichment in processes related to the metabolism of plant growth regulators. Additionally, an increase in the expression of genes involved in key metabolic pathways, such as phenylpropanoid biosynthesis, tryptophan metabolism, and zeatin biosynthesis, is observed. This increase in expression begins with 1DAI and is maintained until the end of the process. Notably, a marked downregulation of genes related to different types of stress is observed. These genes show high activity from 14DBI, but their expression decreases significantly after induction.

The brown module strongly associates with the embryogenic stage and is divided into three distinctive clusters in Figure 3C. Cluster 1 is characterized by genes with low or no expression before 14DAI, which then increases significantly. Interestingly, this group does not present enrichment in specific GO categories or biological pathways. Cluster 2 includes genes that already had considerable expression before 14DAI but increased their activity from this point onwards. This group shows enrichment in processes related to the cell cycle and DNA organization, indicating an intensification of cell division and differentiation during the embryogenic phase. A marked downregulation of chloroplast components distinguishes cluster 3.

Given the unique expression pattern and lack of evident functional enrichment in cluster 1 of the brown module, a more detailed analysis of the genes that comprise it was performed. This group consists of 334 genes with known homologs in *A. thaliana* and 62 without functional annotation in this model organism. Among the identified genes, transcription factors previously associated with SE were found in other species, including four members of the *WOX* family, *BBM*, and *ABI3*. Additionally, genes involved in IAA metabolism were identified, such as *YUC4*, a member of the *GH3* family, and *PIN*, among others.

The STRING platform was used to explore possible interactions between the members of this group. The analysis revealed a leading network of 193 proteins, which was divided into eight clusters using k-means clustering. This network is illustrated in Figure 4. Details on the node degree of the proteins and the composition of the clusters can be found in Appendix A.

The largest cluster in this network is enriched in proteins related to cell division, with MAD2 and CYCA1-1 as central nodes. The second largest cluster includes proteins involved in cutin, suberin, waxes, and fatty acid elongation biosynthesis. The third cluster contains histone components, such as HTR2, and proteins involved in DNA methylation, such as CMT3 and FDM1.

Active cell division is an important feature of in vitro embryogenic cultures; genes controlling the cell cycle and DNA replication are overexpressed during SE in *A. thaliana* [65].

Notably, the fourth cluster groups WOX transcription factors (WOX5, WOX8, and WOX11), CUP2, and proteins related to auxin homeostasis (YUC4, GH3.1, PIN1, LAX2, and IAMT1). Other key transcription factors for forming somatic embryos, such as ABI3 and BBM, are found in the sixth cluster.

Responses mediated by plant growth regulators lead to changes in endogenous auxin levels, which induce the activation of SE marker genes such as *WUS*, *SERK*, *BBM*, *LEC*, and *WOX* [66] in multiple species, such as longan [67], *Camellia sinensis* [68], barley [69], and the embryogenic callus of *C. arabica* [36], among others. In our research, these genes were positively related to the embryogenic stage, when the proembryogenic tissue begins to emerge. These genes were grouped in the brown module along with genes involved in cell cycle control and some *YUC*, *GH3*, *PIN*, and *ARF* family members.

Epigenetic factors regulate gene expression during SE. *WUS*, *LEC1*, *LEC2*, *BBM*, and *WOX* genes and *LATERAL ORGAN BOUNDARIES (LOB) domain* (*LBD*) genes *YUC*, *TAR1*, and *ABI3* are targets of epigenetic regulation [15]. The expression of genes in *A. thaliana* that regulate callus formation in response to wounds, such as *PLETHORA 3* (*PLT3*), *PLT4* and *PLT5*, *WOUND-INDUCED DEDIFFERENTIATION 3* (*WIND3*), and *ETHYLENE RESPONSE FACTOR 115* (*ERF115*), requires H3K27me3 demethylation [70]. During wheat transformation and regeneration, the expression of genes such as *WOX5*, *WUS*, and *LEC2* are associated with dynamic H3K4me3 modifications [71].

### 2.4. Expression of Genes Involved in Auxin and Cytokinin Metabolism During SE of C. canephora

#### 2.4.1. Expression of Auxin Biosynthesis and Signaling Genes

Auxins are involved in multiple plant development processes [72]. In response to auxin perception, gene expression patterns can be affected in a free auxin concentration-dependent manner. For this reason, the homeostasis of this growth regulator is essential.

To analyze the transcriptional role of the genes involved in IAA homeostasis, the genes encoding enzymes related to the metabolism of this hormone were initially selected in *A. thaliana*. Subsequently, the best hit of BLAST results determined the corresponding homologs in *C. canephora*. In total, 172 genes from *A. thaliana* involved in the biosynthesis, conjugation, transport, degradation, and signaling of IAA were selected. In *C. canephora*, 181 genes homologous to those of *A. thaliana* were identified. The complete list of identifiers for both species is available in Appendix A, where it can be observed that not all *A. thaliana* genes have an identified homolog in *C. canephora*. However, it is also observed that some genes in *A. thaliana* have multiple homologs in *C. canephora*, reflecting the evolutionary differences between both species. Additionally, some functional characteristics of the proteins identified in both species were evaluated, including their predicted subcellular localization, the presence of signal peptides, and their classification as soluble or membrane-anchored; this information can be consulted in Appendix A. The results indicated that 77.59% of the proteins presented coincidences in their subcellular localization, suggesting that the genes in *C. canephora* probably perform functions in organelles and cellular compartments like those reported in *A. thaliana*. Regarding the peptide signals, a 71.03% coincidence was observed, reflecting a more significant variability in the signal elements present in the proteins of both species. On the other hand, the properties related to solubility and membrane association showed a concordance of 97.04%, indicating that these structural characteristics are highly conserved or essential for their function in both species.

Significant variations in predicted features were observed when classifying genes according to their role in IAA homeostasis. For example, genes involved in tryptophan biosynthesis showed 100% concordance in the three features analyzed, suggesting a robust functional conservation of these proteins. In contrast, the predicted subcellular localization for the TAA and YUC families showed only 47.05% concordance. Furthermore, the presence of the same signal peptides in the proteins responsible for IAA conjugation was limited to 54.28%.

After identifying the homologous genes and analyzing some of their functional features in silico, the expression patterns of these *C. canephora* genes were evaluated throughout the four defined stages of SE: pretreatment, early response, late response, and embryogenic response. This analysis aims to determine whether any member of the gene families involved in IAA homeostasis plays a key role at specific stages of SE.

Figure 5 provides a detailed view of the expression patterns of genes involved in IAA homeostasis in *C. canephora*. The genes are organized according to the organelles in which they have been localized in *A. thaliana*. Since some libraries present differences in their sequencing depth, TPM values were used to measure expressions to generate heat maps. TPM values were averaged per stage, and genes with TPM equal to 0 in all samples were excluded from the figure.

The complete list of genes, including the size of the encoded protein, TPM values per library, log_2_FC, *p*-adjusted, and the module to which they belong, is available in Appendix A. The color scale used in the heatmaps is uniform for all gene families and varies from light yellow to red to reflect expression levels. Expression values equal to 0 are outside this scale and are represented in white. In addition, the identifiers of some genes were highlighted with colors corresponding to the co-expression module to which they belong. This detail was added only to the genes that are part of the most representative modules in abundance within the genes associated with auxin homeostasis: pink, turquoise, yellow, brown, blue, and red. Genes belonging to other modules are also recorded, and this information can be consulted in detail in Appendix A.

The *C. canephora* genes were named according to their closest homolog in *A. thaliana*. In cases where more than one *C. canephora* gene shared the identical homolog, they were assigned letters alphabetically to differentiate them, respecting their location in the genome. For example, in the case of the *TIR1* gene, four homologs were identified in *C. canephora*, of which one did not show expression at any stage of sampling. In Figure 5, the gene on chromosome 1 was named *TIR1*-a, the one on chromosome 2 was called *TIR1*-b, and the one on chromosome 6 was labeled *TIR1*-c.

The main IAA biosynthesis pathway conserved in angiosperms is tryptophan-dependent [73]. The biosynthesis of tryptophan is carried out in plastids through a series of seven enzymatic reactions. As seen in Figure 5, the genes encoding these enzymes show relatively consistent expression levels throughout the SE process. However, three genes, *ASA1*, *TSA1*, and *TSB4*, stand out for presenting a considerable increase in their expression during the first hours of induction, with *TSB4* exhibiting the most pronounced increase at this early stage. Additionally, as shown in Appendix A, the genes involved in Trp biosynthesis have higher expression levels than the average of all genes in *C. canephora* and the identified genes involved in auxin biosynthesis.

To convert tryptophan into IAA, it must first be transported to the cytosol, where it will be transformed into indole-3-pyruvic acid (IPyA) by tryptophan aminotransferase enzymes of the *TAA* gene family. IPyA is subsequently converted to IAA by flavin-containing monooxygenases of the *YUC* family [74]. The data in Figure 5 show that some genes of these families present different expression patterns than other members of their gene family. In the *YUC* family, the *YUC10*-a gene stands out for being the most expressed throughout all stages of SE, while *YUC4* shows expression only in the embryogenic stage. On the other hand, *YUC10*-b is expressed throughout the process, reaching its maximum expression in the embryogenic stage. Although *YUC* family members catalyze the same enzymatic reaction, our data suggest that *YUC4* and *YUC10*-b, both associated with the brown coexpression module, might play more prominent roles during embryo formation or be restricted to specific tissues. This association and their differences in expression levels and timing suggest that these genes do not have wholly redundant but rather complementary functions during SE. On the other hand, only two of the seven *TAA* genes are expressed throughout the entire process; both decrease their expression considerably during the first hours of induction; then, after 24 h, *TAR4*-b progressively increases its expression. While *TAA* and *YUC* expression decreases during early SE, *ILL4*-a and *ILL6* exponentially increase their expression at this stage. The IRR/IRL family releases IAA conjugated to amino acids, thus restoring its active form [75]. In this sense, the possibility arises that the cells’ need for IAA to respond to the induction stimulus would be supplied by the release of stored inactive IAA instead of depending on Trp, which might be required for other cellular processes.

The IAA-amido synthase *GH3* family is more complex because proteins encoded by members of this family could differ in their specificity toward the amino acids with which they conjugate IAA and their expression patterns throughout SE. Appendix A shows that, in addition to the genes of group II of the *GH3* family, other genes, such as members of the UDP-glycosyltransferase family and *IAMT1*, are involved in IAA conjugation through glucoside formation and methylation, respectively. On the other hand, DAO1 is responsible for the main IAA degradation pathway, which regulates the levels of free IAA [74]. The homologs of these genes in *C. canephora* exhibit a dynamic pattern of expression: a rapid increase at the beginning of induction, followed by a decrease during the first days, and then showing a progressive increase until reaching 28DAI of SE.

In addition to de novo synthesis and conjugation, auxin transporters are essential in generating and maintaining auxin gradients, which are needed to form different plant structures, such as roots, meristematic centers, and even somatic embryos. These transporters are divided into two main groups: efflux transporters and influx transporters [76].

Among the efflux transporters, we can find PIN-FORMED (PIN), ATP-Binding Cassette Subfamily B (ABCB), and some members of ABC subfamily G (ABCG); these transporters move IAA out of the cell. Although PIN-LIKES (PILS) do not export IAA out of the cell, they transport auxins into the lumen of the endoplasmic reticulum (ER), thus regulating the levels of free auxin in the cytoplasm, therefore limiting its availability of IAA for signaling and storage processes.

On the other hand, influx transporters import extracellular IAA into the cytoplasm; among the transporters of this type, we can find Auxin Influx Carrier (AUX/LAX), NITRATE TRANSPORT (NRT), and some members of ABCB can import IAA under certain conditions. Although efflux and influx transporters play opposite roles, their expression patterns show similar dynamics during the early stages of the SE process. A marked decrease in the expression of these transporters is observed in the early stage, followed by a progressive increase in the 1DAI and 3DAI stages, illustrated in Appendix A.

Finally, free auxin is perceived in the nucleus by TIR1/AFB, a protein part of the SKP-CULLIN-F-BOX (SCF) complex [77]. This complex functions as an E3 ubiquitin ligase that marks AUX/IAA repressors for degradation by adding ubiquitins, directing them to the proteasome 26S. The degradation of AUX/IAA allows the release and activation of Auxin Response Factors (ARF) transcription factors, which regulate the expression of specific target genes. Figure 5 and Appendix A show that genes involved in this pathway, such as *TIR1/AFB*, *AUX/IAA*, and *ARF*, are downregulated during the early stage of SE. However, IAA19 expression increases significantly during this stage.

#### 2.4.2. Expression of Cytokinin Biosynthesis and Signaling Genes

After analyzing the genes related to IAA homeostasis, studying the behavior of the genes involved in CK homeostasis, given their interaction with auxins, is essential. CK, such as 6-benzyladenine (BA) used in the induction medium, performs complementary and sometimes antagonistic functions to auxins by modulating their synthesis, transport, and signaling. The genes involved in CK homeostasis were identified using the same procedures applied to analyze auxin-related genes. Due to the limited knowledge in model organisms about aromatic-type CK, only four isoprene-type CKs were considered in this work: trans-zeatin (tZ), cis-zeatin (cZ), isopentenyladenine (iP), and dihydrozeatin (DZ). One hundred-six genes related to CK biosynthesis, transport, degradation, and signaling were identified in *A. thaliana*, and 87 homologous genes were found in *C. canephora*. The complete list of identifiers is presented in Appendix A. Not all *A. thaliana* genes have a homolog in *C. canephora*, while some genes have more than one homolog in the latter species, which could allow for functional diversification of some paralogs.

Bioinformatics analysis of the proteins encoded by these genes revealed that 76.47% of the predicted subcellular locations are conserved between both species. This indicates a predominant conservation in the compartments where these proteins act. On the other hand, the characteristics related to protein solubility reached a concordance of 95.29%, highlighting the importance of this property in function. The complete data of this analysis are available in Appendix A.

Figure 6 shows how genes related to CK homeostasis are expressed in *C. canephora* throughout the embryogenic process. In the analysis, genes are grouped according to the families to which they belong and the subcellular compartments where their activity has been described in *A. thaliana*. Appendix A details complementary information, such as the size of the encoded proteins, individual TPM values, expression changes (log_2_FC), statistical adjustments (*p*-adjust), and the coexpression module to which they belong.

The biosynthesis of isoprene-type CK begins with the transfer of an isopentenyl group to adenine nucleotides or tRNA, a process catalyzed by isopentenyltransferase enzymes (IPTs), which gives rise to CK nucleotides [78]. Cytosolic IPTs are mainly involved in synthesizing *c*Z nucleotides, whereas plastid-localized IPTs produce the precursors of iP, *t*Z, and DZ. In *C. canephora*, four homologs of plastid IPTs and three homologs of cytosolic IPTs were identified; the latter show higher expression levels than plastid IPTs, as illustrated in Figure 6.

iP and *c*Z nucleotides are synthesized directly by the action of IPTs. However, to generate *t*Z and DZ nucleotides, iP nucleotides must be hydroxylated by CYP735A1, an enzyme anchored to the ER membrane. CYP735A1 converts iP nucleotides into tZ nucleotides, which require an additional step to be transformed into DZ nucleotides; currently, the enzyme that carries out this reaction has not been identified. Finally, CK nucleotides can be converted into free bases, active forms of CK, through a reaction catalyzed by LOG enzymes [79]. Figure 6 shows that *C. canephora* presents a single homolog of CYP735A1, whose expression is undetectable during the pretreatment stage. On the other hand, 10 *LOG* homologs were identified, responsible for converting CK nucleotides into active free bases. Among them, the expression of *LOG1-c* stands out, which shows a progressive increase in its expression from the late response stage. LOG5-b exhibits a significant upregulation during the early response, and *LOG7-b* belongs to the brown module associated with the embryogenic response stage.

The adenine phosphate ribosyl transferase (APT) family enzymes catalyze a reaction opposite to LOG, converting active forms into inactive CK nucleotides. The four *APT* homologs identified in *C. canephora*, especially *APT1*, present higher expression levels than LOG.

The degradation of CK is mediated by CK oxidase/dehydrogenase (CKX), which catalyzes the irreversible cleavage of the isoprenoid group [80]. In Figure 6, we can observe that *C. canephora* has five *CKX* genes. Except for *CKX6*, which is predominantly expressed during the early and late response, the rest of the *CKX* genes present a higher expression during the late response. Similar to what was observed in *A. thaliana*, different cellular localizations are predicted for the *C. canephora* CKX members (Appendix A), so they are expected to be found in the cytosol, apoplast, ER, and nucleus.

CK transport is crucial for their function as growth regulators. Families of transporters such as ENT, ABCG, PUP, AZG, and BRITTLE are involved in CK movement. Still, some primarily transport nucleosides or purine bases, representing a challenge in studying CK transport [81].

Finally, CK signaling is carried out through receptor histidine kinases (AHKs) that detect CK and transfer the signal through phosphate transfer proteins (AHPs) to response regulators (ARRs) located in the nucleus. In addition to this central mechanism, cytokinin response factors (CRFs) can interact with some pathway components, such as AHKs, AHPs, and ARRs, modulating gene expression and expanding the complexity of CK signaling [82]. As shown in Figure 6, complemented with the differential data obtained using DESeq2 and the TPM values reported in Appendix A, specific expression patterns are observed in the genes involved in this pathway. For example, *AHK1* is the only member of its family that increases its expression during the early stage of induction, while *AHK4* presents a higher expression during the late response stage. In the case of AHP genes, *AHP1-b* shows initial repression in the early stage, followed by upregulation in the late stage, while *AHP1* maintains constant levels throughout the process. On the other hand, AHP4 presents higher expression in the late stage, while *AHP4-b* progressively reduces its expression after the early stage. These families are essential for sensing and transferring CK signals to the nucleus.

In the nucleus, AHPs phosphorylate ARRs, divided into type A and type B. Type A ARRs act as repressors by interacting with type B ARRs, activating transcription factors. In *C. canephora*, no homologs were identified for 12 ARRs present in *A. thaliana*, where the latter species has 12 type A *ARR* genes (10 typical and two atypical) and 12 type B *ARR* genes. In contrast, *C. canephora* has three complete type-A *ARR* genes and six type-B *ARR* genes. Type A *ARRs* in *C. canephora* show more dynamic expression patterns, while among type B *ARRs*, *ARR11* stands out, which increases its expression in response to pretreatment.

Additionally, *CRFs* can activate gene expression independently of *ARRs*. Four members of this family of transcription factors were identified in *C. canephora*. While *CRF4*-a and *CRF4*-b are upregulated during the early stage, *CRF4*-c and *CRF5* show repression at this same stage.

SE in *M. truncatula* requires the addition of auxin (2,4-D) and cytokinin (BA); adding only BA will produce a few embryos [62]. In *M. truncatula*, *WUS* expression during SE is CK-dependent, whereas in *A. thaliana*, it depends on auxin gradients associated with *PIN1* expression [83,84].

It has been shown that there is cross-talk between auxin and CK signaling [44,85]. This interaction is particularly evident in genes such as type-A *ARR7* and *ARR15*, whose promoters contain response elements for both regulators, allowing their expression to be modulated by both hormonal pathways. CK promotes the transcription of these genes, while auxin, through *ARF5*, acts as a negative regulator [86]. Additionally, auxin signaling can modulate the local concentration of biologically active auxin by promoting the expression of genes from the *GH3* family [87]. Our work identified the differential expression of two Type-A *ARRs* during pretreatment, upregulatedCcan006g024350 (*ARR9*) and downregulatedCcan001g027820 (*ARR17*). Type-A *ARRs* compete with type-B *ARRs* for phosphorylation mediated by histidine-phosphotransporter proteins (AHPs), thus reducing the transcriptional activation of target genes of CK signaling [88]. Consistent with Rashotte et al. [89], type-B *ARRs* did not present significant changes in their expression during pretreatment. However, from 1HAI onwards, most genes involved in CK metabolism presented differential expression at some point. In addition, some members of multigene families were grouped in different co-expression modules.

As can be seen in Appendix A, during induction (from 0DBI onwards), most of the genes involved in auxin and CK metabolism were found to be differentially expressed at least at one stage of the SE process. Most of these genes belong to gene families with numerous members where they could have redundant functions. However, we observed that members of the same family could belong to different co-expression modules. Therefore, they could be under different regulations and be required in specific tissues or moments.

## 3. Materials and Methods

### 3.1. Biological Material and Growth Conditions

*C. canephora* plants from the germination of somatic embryos were used as plant material. They were grown in a maintenance medium consisting of a semisolid medium from Murashige and Skoog (MS) [90] free of plant growth regulators (PGRs) and supplemented with 29.6 μM of thiamine-HCl, 550 μM of myo-inositol, 0.15 μM of cysteine, 16.24 μM of nicotinic acid, l9.72 μM of pyridoxine-HC, 87.64 mM of sucrose, and 0.285% (*w*/*v*) gellan gum under photoperiod conditions of 16/8 h light/dark at 25 ± 2 °C. They were subcultured in fresh medium every 4 weeks. These plants in the maintenance medium without PGRs were used as a control or reference group for subsequent analysis. SE in *C. canephora* has two key stages: preconditioning and induction [37]. The first stage involves transferring micropropagated *C. canephora* plants to a pretreatment medium containing 0.44 µM of NAA (Sigma, St. Louis, MO, USA, N1145) and 2.32 µM of KIN (Sigma, St. Louis, MO, USA, K0753) and cultivating for 14 days under the same photoperiod conditions.

After the pretreatment, the second and third pairs of leaves from donor plants were selected, and 8 mm diameter leaf discs were cut using a sterile punch. These leaf discs were transferred to 250 mL flasks with 50 mL liquid medium to induce SE. This medium consists of modified Yasuda salts [47] supplemented with 5 µM of BA (PhytoTechnology Laboratories, Lenexa, KS, USA, B800). The explants were incubated with shaking (60 rpm) in dark conditions at 25 ± 2 °C for 28 days [37], the time in which the induction of the SE of *C. canephora* ends and the embryonic structures begin to appear.

### 3.2. RNA Extraction and Sequencing

Total RNA was isolated from 60 mg of leaf tissue (between six and seven circular explants) using Plant/Fungi Total RNA Purification Kit with DNase treatment (Norgene, Thorold, ON, Canada. Cat. 25800 and 25710, respectively) according to the manufacturer’s instructions in a final volume of 25 µL. To avoid degradation, all steps were carried out at a low temperature. RNA concentration and integrity were determined using a NanoDrop^TM^ 2000 spectrophotometer (Thermo Scientific, San Jose, CA, USA) and 1.2% agarose gel electrophoresis. For RNA preservation, the RNAstable^®^ Tube Kit (Sigma, 93221-001) was used according to the manufacturer’s instructions. For the preparation and sequencing of the libraries, the service provided by the sequencing company Novogene (Sacramento CA, USA) (https://www.novogene.com/us-en/) was used. RNA-Seq libraries were sequenced on an Illumina Platform to a sequencing depth of at least 40 million read pairs (150 bp paired-end sequencing) per sample. The raw sequencing reads were deposited in the NCBI Sequence Read Archive (SRA) under BioProject accessions PRJNA123021 and PRJNA1232050.

### 3.3. Bioinformatic Analysis

The quality of the sequencing data was assessed with FastQC v0.11.7 [91], and Trimmomatic v0.38 [92] was used to remove low-quality bases and adapters.

The filtered reads were mapped to the reference genome [93] with HISAT2 v2.1.0 [94]. The only change was --max-intronlen 50000, and the mapped reads were counted with the count v0.11.2 [95] with the parameters -t exon -I gene_id -f sam --stranded=no --attr gene_id. The Pearson correlation between biological replicates and between samples, PCA, and PERMANOVA test analysis was conducted in Rstudio 2024.04.1+748. TPM values were obtained using StringTie v2.2.1 [96] in expression estimation mode to compare the expression level of the gene families within each sample.

To identify the expression of genes related to auxin and CK metabolism in *C. canephora*, genes annotated in biological processes that included the words “auxin” or “cytokinin” were subsequently retrieved from Gene Ontology Annotations GOslim of *A. thaliana* (downloaded on 22 August 2024). The *C. canephora* homologs were obtained from previous annotations from the DIAMOND and OrthoFinder results. The heatmap of auxin and CK genes was generated in ggplot2 v3.5.1.

Differential expression was performed using the DESeq2 package v1.44.0 [97] in which all libraries were compared against the day 14DBI (untreated plants), differentially expressed genes were selected with the parameters *p*-adjusted < 0.05, log_2_FC > 1 for upregulated genes, and log_2_FC < −1 for down-expressed genes. These genes’ closest *A. thaliana* homologs were retrieved and uploaded to the DAVID [98] with the default parameters for the GO term enrichment analysis of the DEGs.

Several data filtering and processing steps were followed to construct the co-expression network. Initially, genes with less than ten raw counts in at least one-fifth of the samples were filtered. The “goodGenes” function of the WGCNA library was then used to filter out outlier genes. The TPM values of genes that passed both filters were then obtained. Genes that did not have values > 1 TPM in at least two samples were removed. The remaining values were transformed using log_2_(TPM+1). To obtain the co-expression modules, WGCNA v1.72-5 [99] was used with the following parameters: soft_power = 18 (selected according to the scale independence and mean connectivity graphs), maxBlockSize = 6000, mergeCutHeight = 0.25, randomSeed = 1234, and corType = “Pearson”

The modules were correlated with the stages of the SE process by binarizing the samples into four stages: pretreatment, early response, late response, and embryogenic response. The “moduleTraitCor” and “moduleTraitPvalue” functions of the WGCNA library were applied, which calculate and evaluate the statistical significance of the correlations between gene modules and the established phenotypic characteristics.

Finally, interaction networks were generated using the online platform STRING (accessed 23 September 2024) [100] to visualize and analyze the interactions between the genes within the identified modules.

### 3.4. Identification of Auxin and Cytokinin Metabolism Genes

In previous research, the expression and characteristics of various genes were analyzed using the genome and the annotation of Denoeud et al. [93] as a reference; however, some genes were not correctly annotated, were fragmented, or could not be assigned to 1 of the 11 chromosomes. These characteristics motivated the decision to use the new genome assembly performed by Salojarvi et al. [101], significantly improving the quality and completeness of genomic information. The genome and annotation of the doubled haploid of *C. canephora* were downloaded from the CoGe platform https://genomevolution.org/ (accessed on 15 January 2024) with the accession number 50947. This genome was assembled with long reads, Dovetail Hi-C, and ultra-high-density linkage map [101] representing an improvement to the *C. canephora* genome published by Denoeud et al. [93].

The functional annotation of the *C. canephora* proteins was performed by identifying their homologous genes in *A. thaliana*. For this purpose, a sequence similarity analysis was carried out using the DIAMOND v0.9.25 [102] program with the Blastp option. The protein sequences of *C. canephora* were compared against a database of representative model genes of *A. thaliana* downloaded through the TAIR portal accessed on 8 April 2024, selecting the best hit or alignment for each *C. canephora protein*. In addition, orthologs and paralogs shared between both species were clustered using the OrthoFinder tool v2.2.0 [103]. The subcellular prediction was performed using the online tool Deeploc 2.1 (https://services.healthtech.dtu.dk/services/DeepLoc-2.1/; accessed on 4 November 2024).

Heat maps were generated in Rstudio to visualize the expression of genes involved in auxin and cytokinin metabolism by transforming the TPM values into log_10_(TPM+1). The figures used to represent the organelles were taken from https://BioRender.com, accessed on 23 February 2025.

## 4. Conclusions

SE is a complex process involving numerous changes in the regulation of cellular metabolism. SE in *C. canephora* has been studied using several approaches, including omics. In this work, we use transcriptomics to describe the main global changes identified throughout 10 key process moments. In Figure 7, we propose a model that summarizes the main changes during the SE induction process in *C. canephora*. Adding NAA and KIN during pretreatment promotes cell wall remodeling while preparing the cell for the stress of induction through the expression of genes encoding chaperone proteins involved in oxidative stress. After induction, an early response occurs, characterized by the expression of stress response genes due to mechanical damage of the explants; correct PGR signaling allows the acquisition of totipotency instead of cell death of the explant. After 24 h of induction, a stage of biosynthesis of cellular components begins in preparation for cell proliferation. Finally, structures where embryos will develop emerge at the edges of the explant. Key genes for SE, such as *BBM*, *WOX*, and *LEC1-like*, are detected transcriptionally. After 28 days, induction ends, giving way to embryo differentiation. Genetic editing techniques and mutant complementation in *A. thaliana* could help better characterize some key genes and complement this research work.

## Figures and Tables

**Figure 1 plants-14-01108-f001:**
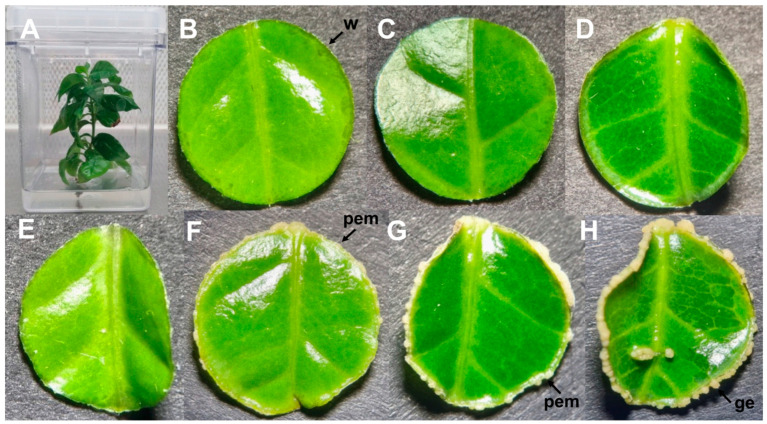
Induction of somatic embryogenesis in *C. canephora*. (**A**) Plant incubated for 14 days in MS preconditioning medium with 0.44 µM of NAA and 2.52 µM of kinetin. Explants at (**B**) one hour, (**C**) two hours, (**D**) one day, (**E**) three days, (**F**) 14 days, (**G**) 21 days, and (**H**) 28 days after being placed in the induction medium. The induction medium is the modified Yasuda medium [47] supplemented with benzyladenine 5 µM. w: wound; pem: proembryogenic mass; ge: globular embryo.

**Figure 2 plants-14-01108-f002:**
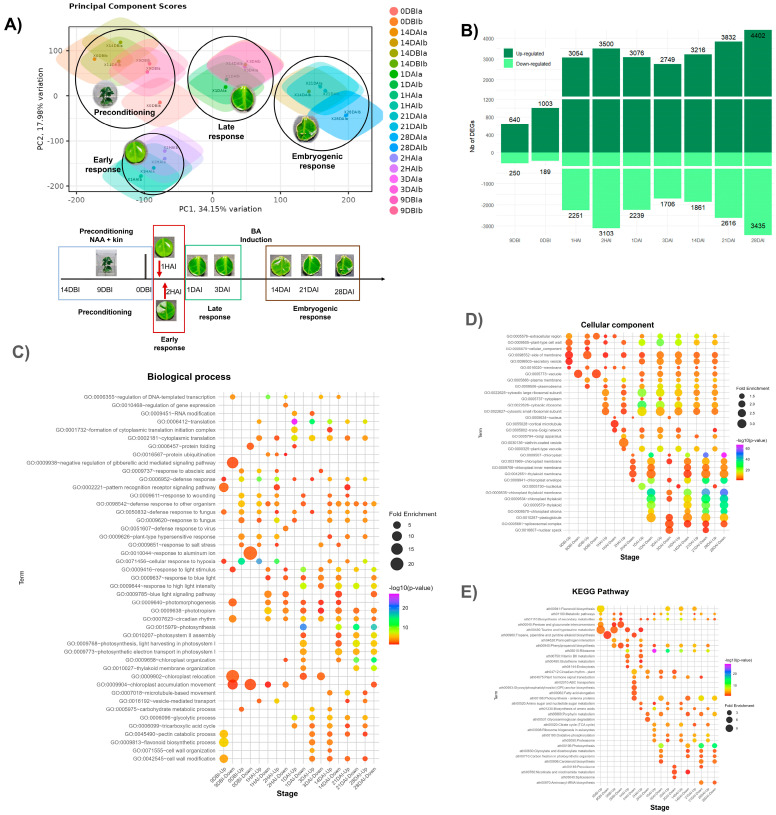
General analysis of the transcriptome of *C. canephora* somatic embryogenesis. (**A**) Principal component analysis (PCA) based on gene expression from 20 RNA-seq libraries. The distribution of samples in space defined by the first two components (PC1 and PC2) reveals four distinct groups, highlighted by black circles, corresponding to pre-treatment, early response, late response, and embryogenic response stages. See lower part of panel A; (**B**) differentially expressed genes (DEGs). (**C**–**E**) Enriched Gene Ontology (GO) terms among DEGs: (**C**) biological process; (**D**) cellular component; (**E**) KEGG pathway.

**Figure 3 plants-14-01108-f003:**
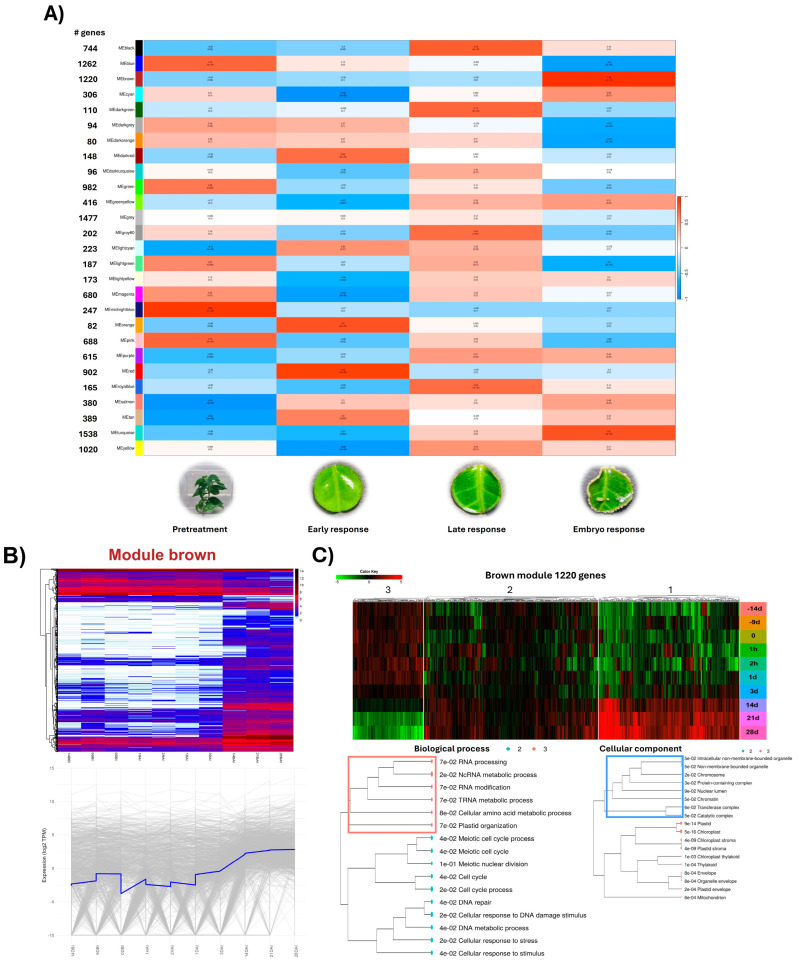
WGCNA analysis of *Coffea canephora* somatic embryogenesis. (**A**) Module–trait association analysis between network modules and the four stages of *C. canephora* SE. The unsigned co-expression network was constructed, and the heatmap displays the correlation values and their respective *p*-values. The number of genes within each module is also indicated. (**B**) Expression patterns of the brown module across SE stages; (**C**) K-means clustering of brown module genes. Gene Ontology (GO) enrichment analysis was performed for the biological process and cellular component categories, while no significant KEGG pathway enrichment was detected. Notably, K-group 1 did not exhibit enrichment in any GO category.

**Figure 4 plants-14-01108-f004:**
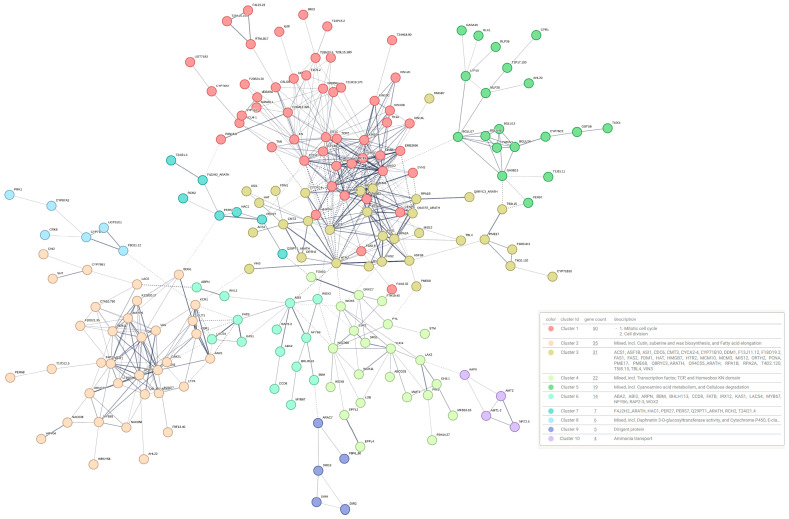
Interaction network of genes from the brown module. The network was clustered using K-means, identifying ten major groups represented by a different color. Functional annotations assigned by STRING for each protein in the network are provided in Appendix A.

**Figure 5 plants-14-01108-f005:**
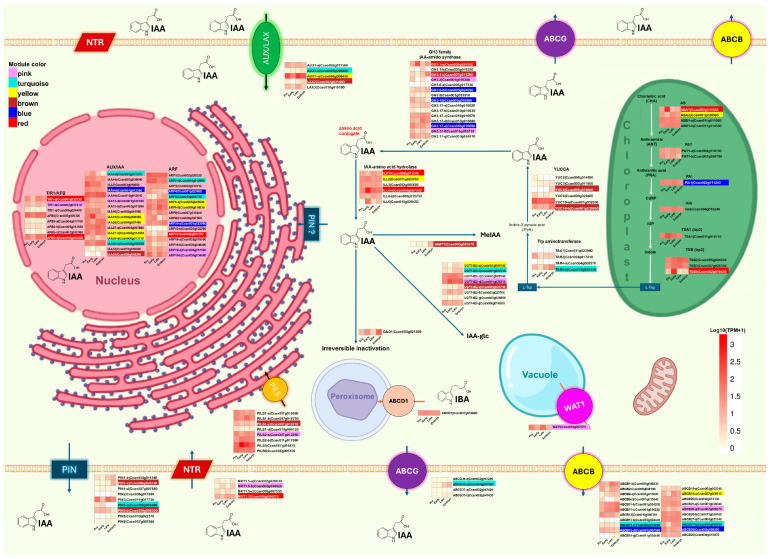
Expression levels of genes involved in IAA metabolism during somatic embryogenesis (SE) in *C. canephora*. The analyzed genes encode enzymes responsible for Trp biosynthesis, IAA biosynthesis and activation, conjugation, degradation, polar transport, and auxin signaling. The spatial representation of organelles is illustrative and not to scale, and gene families are arranged based on their subcellular localization reported for *A. thaliana*. The figures used to represent the organelles were taken from https://BioRender.com (accessed on 23 February 2025).

**Figure 6 plants-14-01108-f006:**
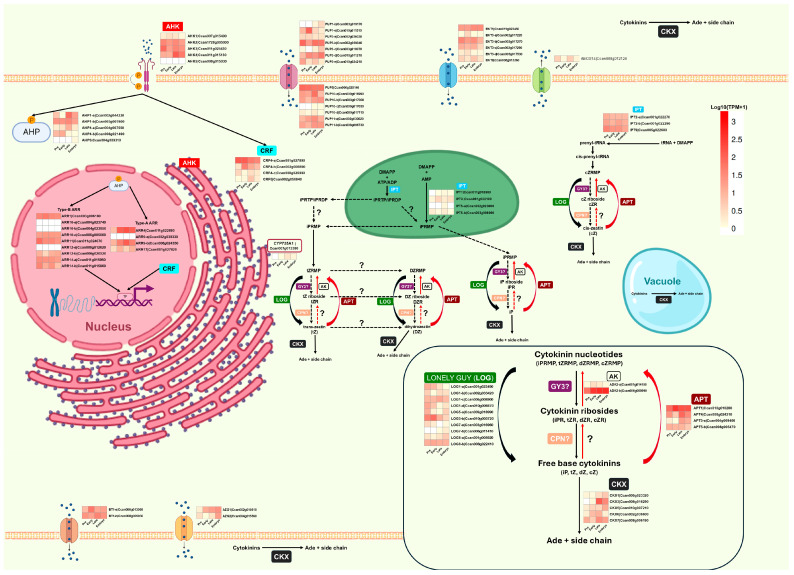
Expression levels of genes involved in isoprene-type CK metabolism during somatic embryogenesis (SE) in *C. canephora*. The analyzed genes encode enzymes responsible for Ck biosynthesis, degradation, transport, and signaling. The spatial representation of organelles is illustrative and not to scale, and gene families are arranged based on their subcellular localization reported for *A. thaliana*. Question marks indicate that the enzymes responsible for the reaction have not yet been identified. The figures used to represent the organelles were taken from https://BioRender.com (accessed on 23 February 2025).

**Figure 7 plants-14-01108-f007:**
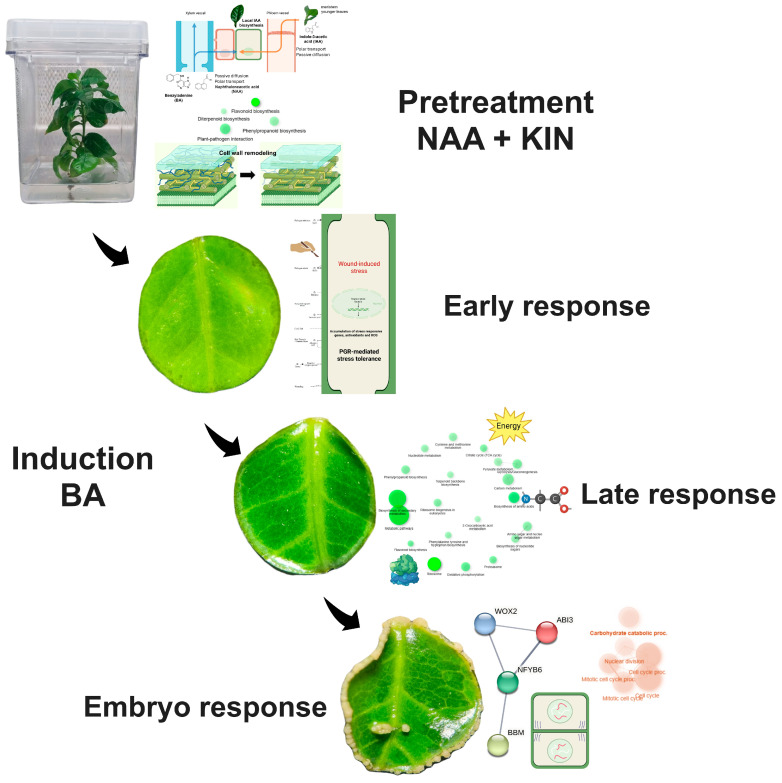
Proposed transcriptomic model for the induction of SE in *C. canephora.*

## Data Availability

The raw sequencing reads were deposited in the NCBI Sequence Read Archive (SRA) under the BioProject accessions PRJNA123021 and PRJNA1232050.

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
