# Peer review of "Transcriptional Dynamics Underlying Somatic Embryogenesis in Coffea canephora"

_plants, 2025, doi:10.3390/plants14071108_

Round 1
Reviewer 1 Report
Comments and Suggestions for Authors
The manuscript presents a detailed transcriptomic study of somatic embryogenesis (SE) in Coffea canephora. The research is significant in plant biotechnology, offering insights into the transcriptional regulation of SE and the roles of auxins and cytokinins in this process. The study utilizes RNA sequencing (RNA-seq) to analyze gene expression at various SE stages. While the manuscript contributes to understanding SE mechanisms, several methodological and analytical aspects require refinement to enhance scientific rigor, clarity, and contextualization.
- Sequencing Depth Variability: The 28 DAI samples were sequenced at significantly lower depth (~19 million reads) compared to other timepoints (~64 million reads). This inconsistency raises concerns about detection sensitivity for late-stage embryogenesis genes.
- Pretreatment Specificity: The pretreatment with NAA/kinetin is described as essential for SE induction, but no control group (e.g., untreated plants) is included. Without this, it is unclear whether observed transcriptional changes are due to pretreatment or inherent stress from explant excision.
- Temporal Resolution Gaps: Sampling intervals (e.g., 1–28 DAI) lack granularity during critical transitions (e.g., 3–14 DAI), potentially missing dynamic gene expression shifts. High-resolution time-series designs, as used in C. arabica SE studies (Nic-Can et al., 2013), could resolve this.
- The manuscript does not sufficiently explain the choice of explant sources, developmental stages, or treatment conditions. A clearer rationale supported by prior studies would improve methodological transparency.
- The study relies exclusively on RNA-seq without independent validation through qRT-PCR. Including a subset of key genes validated by qRT-PCR would strengthen the findings. No qRT-PCR or in situ hybridization validates key differentially expressed genes (e.g., WOX, BBM). Without experimental validation, the transcriptomic findings remain hypothetical.
- The manuscript does not detail the growth conditions of C. canephora plants before explant collection. Given the influence of environmental factors on gene expression, this omission reduces reproducibility.
- Although gene expression patterns are reported, the study does not validate key genes' roles using genetic or biochemical approaches (e.g., overexpression or knockdown studies in model systems).
- The manuscript mentions differential expression analysis but does not provide details on normalization methods, threshold selection for differentially expressed genes (DEGs), or batch effect correction.
- PCA is presented as evidence of transcriptomic changes but lacks a statistical test for cluster separations. Including a PERMANOVA or similar analysis would confirm significant differences between developmental stages. In addition, PCA shows high dispersion among 9DBI and 0DBI replicates (Figure 2A), suggesting biological variability or technical noise. The manuscript does not address whether outlier replicates were retained or excluded, risking skewed conclusions.
- While Gene Ontology (GO) and KEGG pathway analyses are performed, the manuscript does not adequately compare results with existing SE datasets from other plant species.
- Potential Bias in Weighted Gene Co-expression Network Analysis (WGCNA): The manuscript does not address whether the chosen network construction parameters (e.g., soft-thresholding power) are optimal. Sensitivity analyses would improve confidence in the identified modules.
- While variance-stabilizing transformation (VST) was applied, the impact of uneven sequencing depth on downstream analyses (e.g., DESeq2) is not explicitly discussed. Methods like trimmed mean of M-values (TMM) normalization could better address depth disparities.
- The manuscript repeats findings between the results and discussion sections. Condensing overlapping information would streamline the narrative.
- The manuscript would benefit from a more extensive discussion comparing C. canephora SE transcriptomic profiles with those from other species such as Arabidopsis thaliana, Medicago truncatula, or Zea mays.
- Figure 1 lacks labels for critical structures (e.g., wound callus, globular embryos).
- Stage Definitions: The proposed "four transcriptional stages" are inadequately demarcated. A schematic timeline correlating morphological changes (Figure 1) with transcriptional clusters (Figure 2A) would improve clarity.
- Gene Nomenclature: Gene identifiers (e.g., YUCCA, GH3) are not linked to C. canephora genome accessions or orthologs in model systems, limiting cross-study comparisons.
- The manuscript does not discuss potential epigenetic modifications influencing SE, despite existing literature on histone modifications and DNA methylation in embryogenic competence.
- While the manuscript highlights differential regulation of auxin and cytokinin signaling genes, it does not fully explore their interactions or downstream effects on transcription factors regulating embryogenesis.
The manuscript presents an important transcriptomic analysis of C. canephora somatic embryogenesis, but improvements in methodological rigor, data analysis transparency, and functional validation are necessary. Addressing these concerns will enhance the study’s impact and its relevance to plant developmental biology and molecular breeding.
Author Response
Transcriptional Dynamics Underlying Somatic Embryogenesis in Coffea canephora.
Marcos-David Couoh-Cauich, Hugo A. Méndez-Hernández, Rosa M. Galaz-Ávalos, Ana O. Quintana-Escobar, Enrique Ibarra-Laclette, and Víctor M. Loyola-Vargas
Dear Editor
We appreciate the opportunity to resubmit our manuscript, including the reviewers' comments. We kindly thank the reviewers for their time and comments, which we have incorporated into the manuscript and improved the document. We have explained below if we have a different opinion than the reviewer's or if we have made the corresponding changes to the manuscript.
Reviewer 1
The manuscript presents a detailed transcriptomic study of somatic embryogenesis (SE) in Coffea canephora. The research is significant in plant biotechnology, offering insights into the transcriptional regulation of SE and the roles of auxins and cytokinins in this process. The study utilizes RNA sequencing (RNA-seq) to analyze gene expression at various SE stages. While the manuscript contributes to understanding SE mechanisms, several methodological and analytical aspects require refinement to enhance scientific rigor, clarity, and contextualization.
- Sequencing Depth Variability: The 28 DAI samples were sequenced at significantly lower depth (~19 million reads) compared to other timepoints (~64 million reads). This inconsistency raises concerns about detection sensitivity for late-stage embryogenesis genes.
Answer. We thank the reviewer's comments. It has indeed been argued that RNA-Seq studies require sufficient read depth to detect biologically important (differentially expressed) genes and that insufficient sequencing diminishes statistical power, whereas excessive sequencing yields only marginal improvements and incurs unnecessary costs. However, it is also true that several studies have been conducted to evaluate and determine the appropriate depth needed for differential gene expression analysis [1,2]. It has been demonstrated that these concerns have no support when the sequencing depth among the samples analyzed is at least 10 M reads. Ten million reads are enough to detect all genes transcribed under the study condition and obtain a high correlation between the biological replicates [3]. Plus, RNA-seq at this depth can replace other technologies, such as microarrays, even generating more confident results for genes whose transcripts can be present in low abundance and be differentially expressed between different samples that are compared [3].
When comparing two or more samples, the data must be normalized simultaneously to adjust for technical variations, such as sequencing depth [4]. The correct application of normalization methods minimizes biases in gene expression analyses [5,6].
So, despite sequencing depth variability, we show that samples in this study can be compared between them after adequate normalization, as mentioned and demonstrated in section 2.2, line 159.
- Pretreatment Specificity: The pretreatment with NAA/kinetin is described as essential for SE induction, but no control group (e.g., untreated plants) is included. Without this, it is unclear whether observed transcriptional changes are due to pretreatment or inherent stress from explant excision.
Answer. Dear reviewer. Since we published the protocol for inducing somatic embryogenesis (SE) in Coffea canephora [7], we have demonstrated that pretreatment is essential for SE induction. Recently, our group provided a specific analysis of pretreatment effects [8]. Pretreatment is obligatory, and SE is abolished without NAA/kinetin. We appreciate your observation. SE is a complex process; only one study can’t address all aspects.
- Temporal Resolution Gaps: Sampling intervals (e.g., 1–28 DAI) lack granularity during critical transitions (e.g., 3–14 DAI), potentially missing dynamic gene expression shifts. High-resolution time-series designs, as used in C. arabica SE studies (Nic-Can et al., 2013), could resolve this.
Answer. We appreciate your comment, but we respectfully disagree. The SE system in C. canephora has been studied for several years in our group. Over the years, its study has been approached from different perspectives, including the analysis of endogenous growth regulators, primarily auxins and cytokinins. In this regard, we took into account some of the critical days where the most significant change in endogenous concentrations of these regulators is observed (i.e., the first hours and days of induction) [9]. For the work by Nic-Can et al. [10] published in 2013, we sampled every 7 days for epigenetic analysis, and the model was C. canephora. The embryogenic systems for Arabica and Canephora are entirely different, as we found that in the Arabica system, cells secrete phenols that inhibit SE [11].
- The manuscript does not sufficiently explain the choice of explant sources, developmental stages, or treatment conditions. A clearer rationale supported by prior studies would improve methodological transparency.
Answer. Dear reviewer, we appreciate your comment and have included a corresponding explanation in the methodology.
- The study relies exclusively on RNA-seq without independent validation through qRT-PCR. Including a subset of key genes validated by qRT-PCR would strengthen the findings. No qRT-PCR or in situ hybridization validates key differentially expressed genes (e.g., WOX, BBM). Without experimental validation, the transcriptomic findings remain hypothetical.
Answer. This is a valuable comment that we appreciate. We agree that qRT-PCR validation would strengthen the presented results. However, the main focus of this work was to characterize gene expression patterns during somatic embryogenesis, providing a solid basis for the identification of candidate genes involved in this process. We also consider that our RNA-seq experimental design is robust enough to guarantee a valid identification of differentially expressed genes (DEGs), taking care of data normalization and relevant statistical analyses at each step. A total of 13,119 DEG were identified, and we consider all of them to be important, so it would represent a significant amount of time and experimental work if we wanted to perform qRT-PCR of each of these genes for each sampling day used, with their respective replicates. This is not a justification; however, this work lays the groundwork for selecting specific candidate genes to be analyzed in depth in future work involving techniques other than qRT-PCR.
- The manuscript does not detail the growth conditions of C. canephora plants before explant collection. Given the influence of environmental factors on gene expression, this omission reduces reproducibility.
Answer. We appreciate the comment; however, we kindly clarify that the information on plant cultivation before induction is detailed in the first paragraph of section 3.1 of the methodology (lines 893-900).
- Although gene expression patterns are reported, the study does not validate key genes' roles using genetic or biochemical approaches (e.g., overexpression or knockdown studies in model systems).
Answer. We appreciate the reviewer's observation and agree that using the abovementioned tools would be valuable in complementing our research. However, coffee is a perennial agricultural crop whose in vitro growth is not as fast as other model plants (Arabidopsis), which would represent significant limitations for the prompt implementation of such tools. For now, this is a general exploratory study aimed at proposing a model and, subsequently, with the genes of interest selected, exploring gene editing tools. However, this process will take months and even years to standardize and obtain regenerated transformants.
- The manuscript mentions differential expression analysis but does not provide details on normalization methods, threshold selection for differentially expressed genes (DEGs), or batch effect correction.
Answer. Threshold selection for differentially expressed genes is detailed in section 3.3, “differentially expressed genes were selected with the parameters P-adjusted < 0.05, log2FC > 1 for up-regulated genes, and log2FC<-1 for down-expressed genes.”
- PCA is presented as evidence of transcriptomic changes but lacks a statistical test for cluster separations. Including a PERMANOVA or similar analysis would confirm significant differences between developmental stages. In addition, PCA shows high dispersion among 9DBI and 0DBI replicates (Figure 2A), suggesting biological variability or technical noise. The manuscript does not address whether outlier replicates were retained or excluded, risking skewed conclusions.
Answer. We appreciate your recommendation. The replicates from times 9DBI and 0DBI show a Pearson correlation coefficient of 0.93, as demonstrated in Supplementary Figure S3; the other samples exhibit even higher values. Therefore, no samples were eliminated from the analysis. Additionally, we conducted the PERMANOVA test, confirming significant differences between developmental stages (P = 0.001), which statistically validates the separation observed in the PCA analysis. Furthermore, the study indicates that the temporal factor accounts for 83.3% of the transcriptomic variability (R² = 0.833), which statistically supports our interpretation of gene expression changes during the various stages of the studied process.
- While Gene Ontology (GO) and KEGG pathway analyses are performed, the manuscript does not adequately compare results with existing SE datasets from other plant species.
Answer. We appreciate this suggestion to enhance the results. To address this point, we have expanded the discussion to include a detailed comparison between the enriched biological processes and metabolic pathways identified in C. canephora and those reported in studies of SE in A. thaliana, Z. mays, and C. arabica.
- Potential Bias in Weighted Gene Co-expression Network Analysis (WGCNA): The manuscript does not address whether the chosen network construction parameters (e.g., soft-thresholding power) are optimal. Sensitivity analyses would improve confidence in the identified modules.
Answer. The soft power value (soft power = 18) used in our analysis was chosen after assessing the Scale Independence and Mean Connectivity graphs presented in Supplementary Figure S8. This value represents the point at which Scale Independence exceeds 0.8. The decision was based on standard procedures recommended in the literature for constructing gene coexpression networks.
- While variance-stabilizing transformation (VST) was applied, the impact of uneven sequencing depth on downstream analyses (e.g., DESeq2) is not explicitly discussed. Methods like trimmed mean of M-values (TMM) normalization could better address depth disparities.
Answer. VST was exclusively used for Pearson correlation analyses between samples in our study. DESeq2 employs its library-size-based normalization method to normalize differences in sequencing depth to address variations in sample read counts. DESeq2 incorporates specific normalization factors that adjust for these discrepancies in differential expression analysis. While the Trimmed Mean of M (TMM) method could be a valid alternative, it was not utilized in this analysis because many references validate the DESeq2 normalization strategy, effectively handling sequencing depth.
- The manuscript repeats findings between the results and discussion sections. Condensing overlapping information would streamline the narrative.
Answer. We have followed your recommendation and removed the duplicate text from the results and discussion. Thank you for this valuable suggestion.
- The manuscript would benefit from a more extensive discussion comparing C. canephora SE transcriptomic profiles with those from other species such as Arabidopsis thaliana, Medicago truncatula, or Zea mays.
Answer. We have incorporated an extended discussion comparing the transcriptomic profiles obtained in other species.
- Figure 1 lacks labels for critical structures (e.g., wound callus, globular embryos).
Answer. Thank you very much for the suggestion. We have modified the image by adding the appropriate labels.
- Stage Definitions: The proposed "four transcriptional stages" are inadequately demarcated. A schematic timeline correlating morphological changes (Figure 1) with transcriptional clusters (Figure 2A) would improve clarity.
Answer. Thank you very much for the suggestion. To clarify the term, we've introduced a timeline in the lower part of panel A of Figure 2.
- Gene Nomenclature: Gene identifiers (e.g., YUCCA, GH3) are not linked to C. canephora genome accessions or orthologs in model systems, limiting cross-study comparisons.
Answer. Figures 5 and 6 and Supplementary Table S8 specify the gene identifiers and their corresponding C. canephora genome accessions and orthologs in Arabidopsis. These resources provide detailed information that facilitates cross-study comparison.
- The manuscript does not discuss potential epigenetic modifications influencing SE, despite existing literature on histone modifications and DNA methylation in embryogenic competence.
We explore how these modifications regulate key transcription factors involved in SE and in plant regeneration.
- While the manuscript highlights differential regulation of auxin and cytokinin signaling genes, it does not fully explore their interactions or downstream effects on transcription factors regulating embryogenesis.
Answer. We appreciate your comment. We've expanded the corresponding discussion to include several aspects related to the signaling of auxins and cytokinins.
The manuscript presents an important transcriptomic analysis of C. canephora somatic embryogenesis, but improvements in methodological rigor, data analysis transparency, and functional validation are necessary. Addressing these concerns will enhance the study’s impact and its relevance to plant developmental biology and molecular breeding.
Answer. We have incorporated the suggested improvements to enhance the data analysis's methodological rigor and transparency. We are confident that these revisions have improved the manuscript. We agree with you on the importance of functional validation; however, as this is an exploratory transcriptomic analysis, the primary aim is to identify key expression patterns and regulatory networks associated with SE in C. canephora. We believe that the extensive data generated from this study provides a valuable foundation for the future selection and functional characterization of candidate genes involved in this process.
Furthermore, we would like to clarify that the goal of this study is not to optimize somatic embryogenesis for molecular breeding purposes but rather to establish a model that will serve as a platform to investigate further the genetic and regulatory mechanisms involved in somatic embryogenesis in C. canephora and other plant species.
We thank them again for your insightful comments, which have improved our work's quality.
References
- Liu, Y.; Zhou, J.; White, K.P. RNA-seq differential expression studies: more sequence or more replication? Bioinformatics 2013, 30, 301-304, doi:http://doi.org/10.1093/bioinformatics/btt688.
- Imin, N.; Nizamidin, M.; Daniher, D.; Nolan, K.E.; Rose, R.J.; Rolfe, B.G. Proteomic analysis of somatic embryogenesis in Medicago truncatula. Explant cultures grown under 6-benzylaminopurine and 1-naphthaleneacetic acid treatments. Plant Physiol. 2005, 137, 1250-1260, doi:http://dx.doi.org/10.1104/pp.104.055277.
- Wang, Y.; Ghaffari, N.; Johnson, C.D.; Braga-Neto, U.M.; Wang, H.; Chen, R.; Zhou, H. Evaluation of the coverage and depth of transcriptome by RNA-Seq in chickens. BMC Bioinform. 2011, 12, S5, doi:https://doi.org/10.1186/1471-2105-12-S10-S5.
- Bolstad, B.M.; Irizarry, R.A.; Åstrand, M.; Speed, T.P. A comparison of normalization methods for high density oligonucleotide array data based on variance and bias. Bioinformatics 2003, 19, 185-193, doi:https://doi.org/10.1093/bioinformatics/19.2.185.
- Bullard, J.H.; Purdom, E.; Hansen, K.D.; Dudoit, S. Evaluation of statistical methods for normalization and differential expression in mRNA-Seq experiments. BMC Bioinform. 2010, 11, 94, doi:https://doi.org/10.1186/1471-2105-11-94.
- Wagner, G.P.; Kin, K.; Lynch, V.J. Measurement of mRNA abundance using RNA-seq data: RPKM measure is inconsistent among samples. Theory in Biosciences 2012, 131, 281-285, doi:https://doi.org/10.1007/s12064-012-0162-3.
- Quiroz-Figueroa, F.R.; Monforte-González, M.; Galaz-Ávalos, R.M.; Loyola-Vargas, V.M. Direct somatic embryogenesis in Coffea canephora. In Plant Cell Culture Protocols, Loyola-Vargas, V.M., Vázquez-Flota, F.A., Eds.; Humana Press: Totowa, NJ, USA, 2006, pp. 111-117, https://doi.org/10.1385/1-59259-959-1:111.
- Carrillo-Bermejo, E.A.; Brito-Argáez, L.; Galaz-Ávalos, R.M.; Barredo-Pool, F.; Loyola-Vargas, V.M.; Aguilar-Hernández, V. Protein profile changes during priming explants to embryogenic response in Coffea canephora: identification of the RPN12 proteasome subunit involved in the protein degradation. PeerJ 2024, 12, e18372, doi:https://doi.org/10.7717/peerj.18372.
- Ayil-Gutiérrez, B.A.; Galaz-Ávalos, R.M.; Peña-Cabrera, E.; Loyola-Vargas, V.M. Dynamics of the concentration of IAA and some of its conjugates during the induction of somatic embryogenesis in Coffea canephora. Plant Signal Behav. 2013, 8, e26998, doi:https://doi.org/10.4161/psb.26998.
- Nic-Can, G.I.; López-Torres, A.; Barredo-Pool, F.A.; Wrobel, K.; Loyola-Vargas, V.M.; Rojas-Herrera, R.; De-la-Peña, C. New insights into somatic embryogenesis: LEAFY COTYLEDON1, BABY BOOM1 and WUSCHEL-RELATED HOMEOBOX4 are epigenetically regulated in Coffea canephora. PLoS ONE 2013, 8, e72160, doi:https://doi.org/10.1371/journal.pone.0072160.
- Nic-Can, G.I.; Galaz-Ávalos, R.M.; De-la-Peña, C.; Alcazar Magana, A.; Wrobel, K.; Loyola-Vargas, V.M. Somatic embryogenesis: Identified factors that lead to embryogenic repression. A case of species of the same genus. PLoS ONE 2015, 10, e0126414, doi:https://doi.org/10.1371/journal.pone.0126414.

Reviewer 2 Report
Comments and Suggestions for Authors
Dear authors, your work on the transcriptome of the SE process in Coffea canephora is impressive and very wide. I congratulate you. but I have some suggestions for you and I hope you will accept them to improve your manuscript.
1. My general suggestion is to divide the text into sections 2.2, 2.3, and 2.4 in subsections to be easy to read the paper.
2. The other main suggestion is to diminish the text in the figure's description to a minimum. Now the description of the figures is too long and a lot of the text in the description is repeated with the text in the results.
3. Please when DBI, HAI, etc appear for the first time describe their meaning.
4. All abbreviations of genes should be in italics throughout the text.
Some improvements to the text.
line 36 change to : embryo development
line 72 change to: somatic embryo fate
line 149-150 change to: first globular embryos observed
line 121 -Not pretreated plants drastically.....
line 183-184 globular embryos have not been observed. This is in contradiction with Fig. 1 part H.
line 234 to embryogenic tissue
Comments on the Quality of English LanguageEnglish is not my native language, so I improved the text with some changes but they are rather scientific than English.
Author Response
Transcriptional Dynamics Underlying Somatic Embryogenesis in Coffea canephora.
Marcos-David Couoh-Cauich, Hugo A. Méndez-Hernández, Rosa M. Galaz-Ávalos, Ana O. Quintana-Escobar, Enrique Ibarra-Laclette, and Víctor M. Loyola-Vargas
Dear Editor
We appreciate the opportunity to resubmit our manuscript, including the reviewers' comments. We kindly thank the reviewers for their time and comments, which we have incorporated into the manuscript and improved the document. We have explained below if we have a different opinion than the reviewer's or if we have made the corresponding changes to the manuscript.
Reviewer 2
Dear authors, your work on the transcriptome of the SE process in Coffea canephora is impressive and very wide. I congratulate you. but I have some suggestions for you and I hope you will accept them to improve your manuscript.
- My general suggestion is to divide the text into sections 2.2, 2.3, and 2.4 in subsections to be easy to read the paper.
Answer. We appreciate the suggestion to divide sections 2.2, 2.3, and 2.4 into subsections to improve readability. After carefully evaluating the structure and flow of the manuscript, we determined that dividing section 2.4 into subsections was appropriate. We believe that maintaining the other sections as single units preserves logical continuity and avoids disrupting the flow of the manuscript.
- The other main suggestion is to diminish the text in the figure's description to a minimum. Now the description of the figures is too long and a lot of the text in the description is repeated with the text in the results.
Answer. Thank you for your suggestion. We agree with you and have addressed the changes.
- Please when DBI, HAI, etc appear for the first time describe their meaning.
Answer. We agree with the observation. We have already placed the definition in the first time used.
- All abbreviations of genes should be in italics throughout the text.
Answer. We apologize for the error. We have made the changes.
Some improvements to the text.
line 36 change to : embryo development
line 72 change to: somatic embryo fate
line 149-150 change to: first globular embryos observed
line 121 -Not pretreated plants drastically.....
line 183-184 globular embryos have not been observed. This is in contradiction with Fig. 1 part H.
line 234 to embryogenic tissue
Answer. We appreciate your comments, which certainly improved the manuscript. We have taken them into account.

Round 2
Reviewer 1 Report
Comments and Suggestions for Authors
The authors have implemented the necessary corrections and revisions, and the manuscript is suitable for acceptance in its current form.